# Comparative analysis of infertility healthcare utilization before and after insurance coverage of assisted reproductive technology: A cross-sectional study using National Patient Sample data

**Han-Sol Lee[1], Yu-Cheol Lim[2], Dong-Il Kim[3], Kyoung-Sun Park[2], Yoon Jae Lee[2], In-Hyuk Ha [2]\*, Ye-Seul Lee [2]\***

**1** Jaseng Hospital of Korean Medicine, Seoul, Republic of Korea, **2** Jaseng Spine and Joint Research Institute, Jaseng Medical Foundation, Seoul, Republic of Korea, **3** Department of Obstetrics & Gynecology, College of Korean Medicine, Dongguk University Ilsan Oriental Hospital, Goyang, Gyeonggi-do, Republic of Korea

\* yeseul.j.lee@gmail.com (YSL); hanihata@gmail.com (IHH)

**Data Availability Statement:** Patient samples can be obtained via the HIRA website by completing the End User Agreement for Patient Samples.

## Abstract

This study aims to analyze the types and cost of infertility care provided in a clinical setting to examine the changes of healthcare utilization for infertility after the 2017 launch of assisted reproductive technology (ART) health insurance coverage in South Korea. Health Insurance Review Assessment—National Patient Sample data from 2016 and 2018 were analyzed comparatively. Data related to receiving medical service under the International Classification of Diseases 10th revision code N97 (female infertility) or N46 (male infertility) at least once were analyzed, including patients' characteristics and healthcare utilization (type of healthcare facility and treatment approach). Between 2016 and 2018, the percentage of patients aged 30–34 receiving infertility care dropped; the percentages of patients in older age groups increased. The number of female patients remained comparable, whereas the number of male patients increased by 23%. Average visits per patient increased by about 1 day from 2016 to 2018. Total annual infertility care claim cost increased from $665,391.05 to $3,214,219.48; the per-patient annual cost increased from $114.76 to $522.38. The number of claims and cost of treatment and surgery increased markedly, as did the number of claims and cost of gonadotropins. With its focus on health insurance coverage of ART and results demonstrating increases in medical services, medications, cost, and patient utilization, this study reveals the significant effects of national health policies on the treatment, cost, and management of infertility.

## Introduction

Infertility refers to the inability to conceive within one year of normal, unprotected sexual intercourse [1]. While infertility and subfertility are generally viewed as interchangeable,

Patient samples are provided in a closed server, and a fee for the samples may be charged. See https://opendata.hira.or.kr/home.do. This is secondary data statistically sampled from the raw data after the removal of personal and legal-entity-related data. The datasets can be accessed upon request and review to HIRA (https://opendata.hira.or.kr/).

**Funding:** This work was supported by the Ministry of Health and Welfare through the Korean Medicine Innovative Technology R&D project managed by the Korea Health Industry Development Institute (HF21C0028). The funders had no role in designing or developing the study protocol; in the collection, analyses, or interpretation of data; in the writing of the manuscript; or in the decision to publish the results.

**Competing interests:** The authors have declared that no competing interests exist.

infertility is used primarily in the assessment and diagnosis of disease. Moreover, a diagnosis of infertility does not refer to "sterility," a state of permanent infertility. The term "infertility" is used throughout the manuscript [2].

The causes of infertility in women include ovulatory dysfunction; fallopian tube disorders; and uterine-related factors, such as endometriosis, uterine fibroids, and congenital uterine abnormality [3]. Ovulatory dysfunctions, such as anovulation and oligo-ovulation, account for 21% of all causes of female infertility [4]. The causes of male infertility can be divided into four broad categories: spermatogenic failure, sperm transport issues, systemic or endocrine disorder leading to hypogonadism, and idiopathic male infertility [5]. Approximately 30–40% of infertility cases are of unknown cause, in which abnormalities cannot be detected by standard tests such as hysterosalpingography and semen testing [6].

A woman's age, the duration of infertility, and the number of previous treatment sessions must be accounted for in infertility care, along with the primary cause of infertility. Ovulatory dysfunctions caused by hypogonadism are treated with gonadotropins. Ovulatory dysfunctions caused by polycystic ovary syndrome can be treated with weight loss and combination therapy of medications using the anti-estrogen drug clomiphene with insulin-sensitive agonist metformin or with assisted reproductive technology (ART). It may be possible to treat a uterine structural problem with surgical correction [7], but in vitro fertilization (IVF) should be considered if there is a fallopian tube problem [3]. In males, primary testicular failure, including oligozoospermia, is the most common cause of infertility, but there is no established treatment [8]; in such cases, artificial insemination or IVF can be considered [9]. For sperm transport issues such as obstructive azoospermia, epididymal obstruction can be corrected surgically [10]; hypogonadism can be treated by injecting gonadotropin [11]. For infertility of unknown cause, the recommendation to patients is to try conceiving naturally for two years before attempting IVF and embryo transfer [12, 13].

The prevalence of infertility is approximately 8–12% in couples of childbearing age worldwide [3]. According to the Centers for Disease Control and Prevention in the United States, approximately 6.7% of married women of childbearing age (15–49 years) were diagnosed with infertility [14]. In South Korea, the number of individuals diagnosed with infertility has risen consistently over the years, from 120,000 in 2004 to 220,000 in 2017 [15]. Total insurance payments for infertility care in South Korea rose steadily from 19.9 billion Korean won (KRW) in 2010 to 33.2 billion KRW in 2017. After expansion of insurance coverage to ART, total insurance payments rose by approximately 3.75 times over a year to 124.6 billion KRW in 2018 [16]. South Korea was ranked as having the lowest total fertility rate (TFR; 0.81) among Organization for Economic Co-operation and Development countries in 2021 [17]. The fertility rate is declining owing to increased later marriages and infertility prevalence, contributing to the population decline.

In recognition of the elevated prevalence of infertility and the gravity of the associated financial burden, the South Korean government began to provide insurance coverage for infertility diagnostic care and testing in 2001. ART was not covered by health insurance at that time, as it was not perceived as life-affecting treatment. However, the government implemented policies and support projects over time to alleviate the financial burden of infertile individuals to increase their access to ART. In 2006, the South Korean government launched the "infertility support project" to provide financial assistance for IVF; a project to provide financial assistance for artificial insemination has been in existence since 2010. In Korean medicine (KM), the number of local governments participating in the KM treatment support project has been increasing continuously. In October 2017, ART was included in health insurance coverage; women aged 44 years and younger are provided insurance coverage for four

rounds of fresh embryo transfer and three rounds of frozen embryo transfer as part of IVF treatment, as well as up to three rounds of artificial insemination.

Insurance coverage of infertility-related medical services can impact both healthcare utilization by consumers and health outcomes, including childbirth [18]. In Taiwan, the government provided financial assistance for ART for the first time in 2007 with the enactment of the Artificial Reproduction Act. A study analyzed the effect of that health policy, finding a 157.8% relative increase in ART treatment frequency three years after the enactment and a 78.51% relative increase in the number of births by ART five years after the enactment [19]. In Germany, ART had been fully covered by the statutory health insurance until December 31, 2003, but only 50% of the cost was covered after that date; since then, the frequency of infertility treatment dropped by 55%, and childbirth by ART dropped by 51% in 2005 compared to 2002 [20]. In South Korea, a 2009 study analyzed the frequency of ART procedures [21]. Another study assessed childbirth after a diagnosis of infertility and the socioeconomic characteristics of infertile patients using Health Insurance Review and Assessment Service-National Patient Sample (HIRA-NPS) data from 2005–2013 [22]. However, none of these studies performed a comprehensive review of changes in healthcare utilization for infertility in both Western medicine (WM) and KM since the inclusion of ART in health insurance coverage in South Korea.

Changes in clinical practice since the inclusion of ART in the health insurance payment system in South Korea in 2017 should be examined through a review of data related to infertility medical practices, medications, and cost of care by service category. In addition, the financial and social burden of infertility before and after health insurance coverage, including total cost of care and out-of-pocket (OOP) cost, should be assessed. This study analyzed the clinical practice and cost of care for patients who utilized healthcare for infertility treatment in 2016 and 2018—before and after health insurance coverage included ART, respectively—using HIRA-NPS data. The study's goal was to find out the effect of the national health policy and provide information for establishing infertility-supportive policies and decision-making on insurance coverage of ART.

## Methods

### Data source

This cross-sectional study used HIRA-NPS data from 2016 and 2018. The HIRA-NPS data contain claims data generated during the process of reimbursing healthcare providers under the National Health Insurance System (NHIS). The HIRA-NPS data are sampled annually by stratifying a random 3% sample (about 1.4 million people) of the entire population of South Korea (individuals registered with NHI or Medical Aid) by sex and age. This is secondary data statistically sampled from the raw data after the removal of personal and legal-entity-related data [23]. The datasets can be accessed upon request and review to HIRA(https://opendata. hira.or.kr/). The results from this study can be replicated by following the protocol in our Methods section. The authors confirm that no privileges of any kind were granted before, during, or after the analysis.

### Study design and population

Patients who received WM or KM services with International Classification of Diseases 10th revision (ICD-10) code N97 (female infertility) or N46 (male infertility) at least once in 2016 or 2018 were included in the study. The identified patients' reimbursement claims data with N97 or N46 as the primary diagnosis were collected for the data analysis. Exclusion criteria were data from patients with claims indicating psychiatric nursing hospital, dental hospital, maternity center, or public health facility as the type of medical institution; patients with

claims in which the total cost or number of visits was 0 or missing; and patients younger than 30 years.

## Study outcomes and analysis

The frequency and percentage of age, sex, payer type, type of visit, and medical institution for patients with infertility were analyzed for the years 2016 and 2018. Age was divided into four groups of five years over the range of 30 to 45 years or older; payer type was divided into NHI, Medicaid, and other. The age criterion was set considering that the use of ART markedly increases over age 30 [21] and that the age of patients who use ART is rising gradually [24, 25]. Type of visit was classified as outpatient or inpatient, and medical institution was classified as tertiary/secondary/primary hospital, clinic, KM hospital, or KM clinic.

Total patients, total cases, total expenses, per-patient expenses, per-case expenses, total visits, and average visits per patient with infertility as the primary diagnosis were analyzed for the entire sample and by sex for the years 2016 and 2018.

High-frequency (top 20) comorbidities, excluding infertility, with reference to the Korean Standard Classification of Diseases (KCD) were analyzed for both sexes. KCD coding is the Korean standard disease classification system for high-frequency diseases in Korea. It was developed based on the International Classification of Diseases published by the World Health Organization.

Service codes for infertility care (limited to codes that accounted for more than 0.1% of the total cases) were classified for both sexes. Total cases, total patients, total costs, annual cost per case, and annual cost per patient were analyzed by service code. Total cost was defined as the sum of the NHIS payment to providers and patients' OOP cost: the allowable amount from the total amount claimed.

After classifying medications prescribed to treat infertility by pharmacies and hospitals into 15 categories according to the Anatomical Therapeutic Chemical Classification System (ATC code), the frequency and cost of prescriptions of each category were compared between 2016 and 2018. Classification of medication is presented in S1 Table.

All costs in this study were converted to the 2018 average South Korean won to US dollar exchange rate and corrected to reflect the consumer price index in the health sector (S2 Table). Data were analyzed using SAS 9.4 software (2002–2012 by SAS Institute Inc., Cary, NC, USA).

## Ethical statement

The study's protocol was approved by the public data provision deliberation committee of the HIRA and was performed in accordance with relevant guidelines and regulations. The current study was reviewed and qualified with an exemption by the Institutional Review Board of Jaseng Hospital of Korean Medicine, Seoul, Korea (JASENG 2021-10-021). As this study analyzed publicly available data, no consent was obtained from participants; all personal information was de-identified by the NHIS prior to public release. The principles expressed in the Declaration of Helsinki were adhered to in the study's analysis.

## Results

A total of 68,091 claims with female infertility (N97) or male infertility (N46) as the primary diagnosis were submitted in 2016 and 2018. After excluding two cases with a medical institution coded as psychiatric nursing hospital, dental hospital, maternity center, or public health facility; 60 cases with total cost or number of visits recorded as 0 or missing; and 9,528 cases of patients under the age of 30 years, 58,501 cases for 11,951 patients were analyzed (Fig 1).

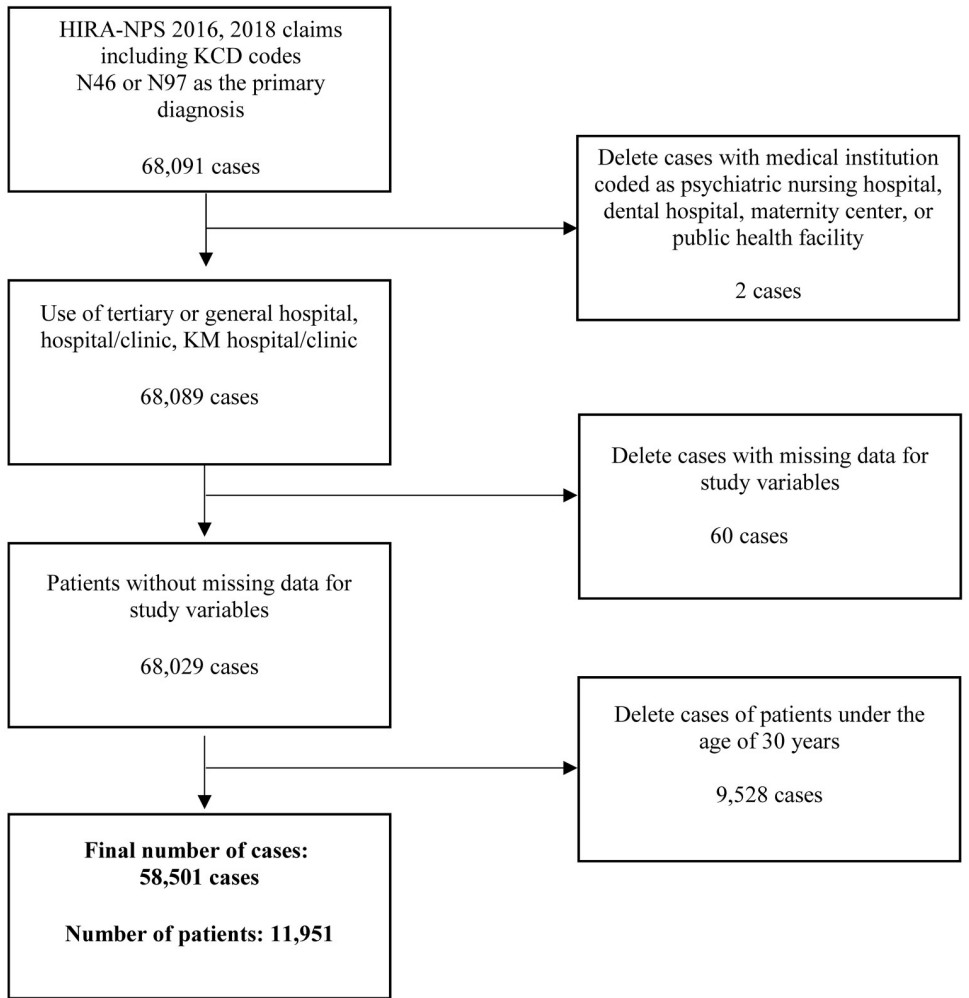

**Fig 1. Flowchart of the selection of the study population.** HIRA—NPS = Health Insurance Review Assessment—National Patient Sample; KCD = Korean Standard Classification of Diseases; KM = Korean medicine.

In both 2016 and 2018, healthcare utilization for infertility tended to decline with advancing age. However, compared to 2016, in 2018 the percentage of patients aged 30–34 years declined, while the percentages in other age groups increased. Among women, the 30–34 years group was significantly the largest patient group in 2016, but the gap between the 30–34 years and the 35–39 years groups narrowed substantially in 2018. Among men, the 30–34 years group was the largest patient group in 2016, but the 35–39 years group was the largest in 2018.

The percentage of women was notably higher than men in both 2016 and 2018. However, while the number of female patients decreased slightly, the number of male patients increased considerably, resulting in an increased percentage of male patients in the infertility patient population. The NHIS was the payer in most cases in 2016 and 2018 (Table 1).

Regarding the number of claims, most patients who presented to a healthcare facility with infertility in both 2016 and 2018 utilized outpatient services, with only a small percentage of patients receiving inpatient services. The most frequently visited medical institution was a clinic; the percentage of patients utilizing a clinic increased slightly in 2018 compared to 2016. Visit type trends were similar across sex and total study sample. While healthcare facility

**Table 1. Characteristics of patients in the study sample.**

| Category | | Total | | | | Female infertility | | | | Male infertility | | | |
|---|---|---|---|---|---|---|---|---|---|---|---|---|---|
| | | 2016 | | 2018 | | 2016 | | 2018 | | 2016 | | 2018 | |
| | | No. of patients | Percent | No. of patients | Percent | No. of patients | Percent | No. of patients | Percent | No. of patients | Percent | No. of patients | Percent |
| Age (years) | 30–34 | 2,824 | 48.71 | 2,536 | 41.22 | 2,102 | 52.51 | 1,808 | 45.85 | 722 | 40.22 | 728 | 32.94 |
| | 35–39 | 1,996 | 34.43 | 2,443 | 39.7 | 1,330 | 33.23 | 1,515 | 38.42 | 666 | 37.1 | 928 | 41.99 |
| | 40–44 | 782 | 13.49 | 938 | 15.24 | 482 | 12.04 | 536 | 13.59 | 300 | 16.71 | 402 | 18.19 |
| | $\geq$ 45 | 196 | 3.38 | 236 | 3.84 | 89 | 2.22 | 84 | 2.13 | 107 | 5.96 | 152 | 6.88 |
| Sex | Male | 1,795 | 30.96 | 2,210 | 35.92 | - | - | - | - | 1,795 | 100 | 2,210 | 100 |
| | Female | 4,003 | 69.04 | 3,943 | 64.08 | 4,003 | 100 | 3,943 | 100 | - | - | - | - |
| Payer type | NHI | 5,775 | 99.6 | 6,141 | 99.8 | 3,987 | 99.6 | 3,938 | 99.87 | 1,788 | 99.61 | 2,203 | 99.68 |
| | Medicaid | 22 | 0.38 | 12 | 0.2 | 16 | 0.4 | 5 | 0.13 | 6 | 0.33 | 7 | 0.32 |
| | Other[a] | 1 | 0.02 | - | - | - | - | - | - | 1 | 0.06 | - | - |

NHI: National Health Insurance;

[a]Veteran health service = 1 case

utilization trends were nearly identical between the female sample and the total sample, male patients barely visited a KM clinic (Table 2).

In 2016, the most common comorbidities of infertility in women were gastritis and duodenitis, followed by gingivitis and periodontal disease, acute bronchitis, and vasomotor and allergic rhinitis. In 2018, the most common comorbidities were gastritis and duodenitis, followed by gingivitis and periodontal disease, childbirth management, and vasomotor and allergic rhinitis (S3 Table). In men, in 2016 the most common comorbidities of infertility were gastritis and duodenitis, followed by gingivitis and periodontal disease, acute bronchitis and vasomotor and allergic rhinitis. In 2018, the most common comorbidities of infertility in men were gastritis and duodenitis, followed by gingivitis and periodontal disease, childbirth management, and vasomotor and allergic rhinitis (S4 Table). In both men and women in 2016 and 2018, gastritis and duodenitis were the most common comorbidities of infertility, followed by gingivitis and periodontal disease, ranked second, and vasomotor and allergic rhinitis, ranked fourth. The third-most common comorbidity of infertility was acute bronchitis in 2016, but the third-most common comorbidity was childbirth management in 2018 among both men and women.

**Table 2. Utilization of medical care for infertility by visit type, location, and sex.**

| Category | | Total | | | | Female infertility | | | | Male infertility | | | |
|---|---|---|---|---|---|---|---|---|---|---|---|---|---|
| | | 2016 | | 2018 | | 2016 | | 2018 | | 2016 | | 2018 | |
| | | No. of cases | Percent | No. of cases | Percent | No. of cases | Percent | No. of cases | Percent | No. of cases | Percent | No. of cases | Percent |
| Type of visit | Outpatient | 23,812 | 99.84 | 34,602 | 99.86 | 21,127 | 99.91 | 30,564 | 99.9 | 2,685 | 99.37 | 4,038 | 99.53 |
| | Inpatient | 37 | 0.16 | 50 | 0.14 | 20 | 0.09 | 31 | 0.1 | 17 | 0.63 | 19 | 0.47 |
| Medical institution | Tertiary/secondary/ primary hospital | 9,874 | 41.4 | 13,845 | 39.95 | 8,335 | 39.41 | 11,849 | 38.73 | 1,539 | 56.96 | 1,996 | 49.2 |
| | Clinic | 12,196 | 51.14 | 18,878 | 54.48 | 11,052 | 52.26 | 16,840 | 55.04 | 1,144 | 42.34 | 2,038 | 50.23 |
| | KM[a] hospital | 367 | 1.54 | 227 | 0.66 | 352 | 1.66 | 211 | 0.69 | 15 | 0.56 | 16 | 0.39 |
| | KM clinic | 1,412 | 5.92 | 1,702 | 4.91 | 1,408 | 6.66 | 1,695 | 5.54 | 4 | 0.15 | 7 | 0.17 |

[a]KM = Korean medicine.

**Table 3. General utilization of medical care for infertility.**

| Year | Sex | Number of patients | Total cases | Total expenses | Per-patient expense | Per-case expense | Total visits | Average visits per patient |
|------|-----|-------------------|-------------|----------------|---------------------|------------------|--------------|---------------------------|
| 2016 | Total | 5,798 | 23,849 | $ 665,391.05 | $ 114.76 | $ 27.90 | 23,921 | 4.13 |
|      | Female | 4,003 | 21,147 | $ 556,803.18 | $ 139.10 | $ 26.33 | 21,193 | 5.29 |
|      | Male | 1,795 | 2,702 | $ 108,587.87 | $ 60.49 | $ 40.19 | 2,728 | 1.52 |
| 2018 | Total | 6,153 | 34,652 | $ 3,214,219.48 | $ 522.38 | $ 92.76 | 34,742 | 5.65 |
|      | Female | 3,943 | 30,595 | $ 2,958,315.79 | $ 750.27 | $ 96.69 | 30,658 | 7.78 |
|      | Male | 2,210 | 4,057 | $ 255,903.69 | $ 115.79 | $ 63.08 | 4,084 | 1.85 |

All expenses were converted with an annual average exchange rate (KRW/USD, see S3 Table); Total visits = number of days in outpatient or inpatient care; Average visits = annual average number of days in outpatient or inpatient care

Acute bronchitis did not appear in the top 20 comorbidities of infertility in 2018; childbirth management entered the top 20 comorbidities in both men and women in 2016.

Regarding the number of claims, most patients who presented to a healthcare facility with infertility in both 2016 and 2018 utilized outpatient services, with only a small percentage of patients receiving inpatient services. The most frequently visited medical institution was a clinic; the percentage of patients utilizing a clinic increased slightly in 2018 compared to 2016. Visit type trends were similar across sex and total study sample. While healthcare facility utilization trends were nearly identical between the female sample and the total sample (Table 2). The number of infertility treatment cases rose by 1.4 times. Total cost of care rose by 4.8 times and per-patient expense rose by 4.5 times. Average visits per patient increased by about 1 day. The average visits per patient gap between the sexes widened from 3.7 days to 5.9 days (Table 3).

The number of cases and the costs for services claimed for infertility in 2016 and 2018 were analyzed for the following services: injection, examination, test, medication/drug-related, treatment and surgery, special equipment and diagnostic radiology, KM acupuncture and electroacupuncture, and KM other. In 2016 and 2018, the three most frequently claimed service codes for both female and male patients were examinations, tests, and injections, and the least frequent service code, treatment and surgery, rose markedly (Tables 4 and 5). The number of

**Table 4. High-frequency service codes for female patients.**

| Category | 2016 | | | | | 2018 | | | | |
|----------|------|------|------|------|------|------|------|------|------|------|
|          | Total cases | Total patients | Total costs | Annual cost per case | Annual cost per patient | Total cases | Total patients | Total costs | Annual cost per case | Annual cost per patient |
| Examinations | 28,451 | 3,925 | $198,375.79 | $6.97 | $50.54 | 42,889 | 3,920 | $315,820.58 | $7.36 | $80.57 |
| Tests | 19,815 | 1,960 | $112,095.20 | $5.66 | $57.19 | 37,664 | 2,472 | $236,932.33 | $6.29 | $95.85 |
| Injections | 3,200 | 1,091 | $7,212.83 | $2.25 | $6.61 | 10,589 | 1,773 | $66,513.59 | $6.28 | $37.51 |
| KM[a] acupuncture and electroacupuncture | 3,015 | 149 | $11,888.87 | $3.94 | $79.79 | 4,389 | 185 | $17,783.17 | $4.05 | $96.13 |
| Medication/drug-related | 2,824 | 1,524 | $302.38 | $0.11 | $0.20 | 7,418 | 2,104 | $861.50 | $0.12 | $0.41 |
| KM other | 2,011 | 118 | $3,846.43 | $1.91 | $32.60 | 2,211 | 168 | $5,452.40 | $2.47 | $32.45 |
| Special equipment and diagnostic radiology | 1,406 | 1,147 | $87,339.61 | $62.12 | $76.15 | 7,508 | 1,859 | $290,598.79 | $38.71 | $156.32 |
| Treatment and surgery | 312 | 269 | $1,458.57 | $4.67 | $5.42 | 4,348 | 1,034 | $1,137,244.61 | $261.56 | $1,099.85 |

All expenses were converted with annual average exchange rate (KRW/USD), see S2 Table;
[a]KM = Korean medicine.

**Table 5. High-frequency service codes for male patients.**

| Category | 2016 | | | | | 2018 | | | | |
|---|---|---|---|---|---|---|---|---|---|---|
| | Total cases | Total patients | Total costs | Annual cost per case | Annual cost per patient | Total cases | Total patients | Total costs | Annual cost per case | Annual cost per patient |
| Tests | 6,416 | 1,483 | $53,194.69 | $8.29 | $35.87 | 10,273 | 1,666 | $87,773.34 | $8.54 | $52.69 |
| Examinations | 4,553 | 1,742 | $31,176.33 | $6.85 | $17.90 | 5,534 | 2,035 | $46,773.07 | $8.45 | $22.98 |
| Injections | 32 | 21 | $43.44 | $1.36 | $2.07 | 44 | 25 | $78.11 | $1.78 | $3.12 |
| Treatment and surgery | - | - | - | - | - | 954 | 658 | $94,356.16 | $98.91 | $143.40 |

All expenses were converted with annual average exchange rate (KRW/USD, see S2 Table).

cases rose in all service categories, including KM-related services, although KM healthcare facilities have relatively decreased in the proportion of medical institution utilization.

Among women, in 2016, examinations were the highest in the cost per service code, followed by tests, special equipment and diagnostic radiology, KM acupuncture and electroacupuncture, and injections. In 2018, treatment and surgery ranked the highest in the cost per service code, followed by examinations, special equipment and diagnostic radiology, tests, and injections. The cost for treatment and surgery rose dramatically, becoming the service code with the highest total cost and annual per-patient cost in 2018 (Table 4).

Among men, in 2016, tests were the highest in the cost per service code, followed by examinations and injections. In 2018, treatment and surgery ranked the highest in the cost per service code, followed by tests, examinations, and injections. There were zero claims submitted for treatment and surgery in 2016, but this service code incurred the highest total cost and annual per-patient cost in 2018. Other service codes, including medication/drug-related, special equipment and diagnostic radiology, KM acupuncture and electroacupuncture, and KM other, did not account for 0.1% of all infertility-related claims in men (Table 5).

High-frequency drug usage ($\geq$ 0.1% of all prescriptions for infertility) and the associated costs were analyzed for the total sample, including both female and male patients. In 2016, ovulation stimulants were the most frequently prescribed, followed by gonadotropins and systemic antibacterials. In 2018, gonadotropins were the most frequently prescribed, followed by systemic antibacterials and ovulation stimulants. In 2016, the total cost for gonadotropins was highest, followed by contrast media and systemic antibacterials. In 2018, the total cost for gonadotropins was highest, followed by systemic hormones and endocrine therapy. Gonadotropins incurred the highest annual per-case and per-patient cost in both 2016 and 2018 (Table 6). Analyzed by sex (only for drugs representing $\geq$ 0.1% of all prescriptions for infertility for each sex), this overall trend of drug usage in total study samples was nearly identical among women (S5 Table). In men, systemic antibacterials were the most frequently prescribed drugs and incurred the highest total annual cost in both 2016 and 2018. However, the number of prescriptions of gonadotropins rose markedly and incurred the highest per-patient cost in 2018 (S6 Table).

## Discussion

This study shows the proportion of women ages 30–34 utilizing infertility healthcare decreased while the proportions of older age groups increased in 2018 compared to 2016. A French study analyzing health insurance data in France reported that the mean age of women utilizing infertility care rose by 0.7 years, from 33.0 years to 33.7 years, between 2008 and 2017, with the percentage of women aged 34 years or older increasing by 23.9% during the period [26]. In Japan,

**Table 6. High-frequency medications for infertility patients, including inpatient and outpatient services and male and female patients.**

| Category | 2016 | | | | | 2018 | | | | |
|---|---|---|---|---|---|---|---|---|---|---|
| | No. of prescriptions | No. of patients | Total cost | Annual cost per prescription | Annual cost per patient | No. of prescriptions | No. of patients | Total cost | Annual cost per prescription | Annual cost per patient |
| Ovulation stimulants, synthetic | 1,873 | 1,063 | $1,895.56 | $1.01 | $1.78 | 1,755 | 1,064 | $1,944.95 | $1.11 | $1.83 |
| Gonadotropins | 1,872 | 603 | $21,273.59 | $11.36 | $35.28 | 6,768 | 1,536 | $392,209.48 | $57.95 | $255.34 |
| Antibacterials for systemic use | 1,516 | 1,055 | $3,083.45 | $2.03 | $2.92 | 2,332 | 1,357 | $4,568.63 | $1.96 | $3.37 |
| X-ray contrast media, iodinated | 1,140 | 1,121 | $12,024.11 | $10.55 | $10.73 | 1,134 | 1,118 | $13,405.37 | $11.82 | $11.99 |
| Musculoskeletal system drugs | 797 | 628 | $665.37 | $0.83 | $1.06 | 1,206 | 859 | $1,060.17 | $0.88 | $1.23 |
| Anti-infectives and antiseptics, excl. combinations | 479 | 323 | $153.80 | $0.32 | $0.48 | 446 | 278 | $188.79 | $0.42 | $0.68 |
| Drugs for functional gastrointestinal disorders | 450 | 388 | $250.55 | $0.56 | $0.65 | 504 | 416 | $334.71 | $0.66 | $0.80 |
| Blood substitutes and perfusion solutions | 309 | 176 | $414.05 | $1.34 | $2.35 | 1,544 | 696 | $1,915.71 | $1.24 | $2.75 |
| Sex hormones and modulators of the genital system | 273 | 147 | $1,093.90 | $4.01 | $7.44 | 891 | 445 | $5,459.59 | $6.13 | $12.27 |
| Others | 260 | 193 | $188.37 | $0.72 | $0.98 | 805 | 497 | $315.39 | $0.39 | $0.63 |
| Anesthetics, analgesics, psycholeptics | 194 | 94 | $381.84 | $1.97 | $4.06 | 1,419 | 606 | $2,176.68 | $1.53 | $3.59 |
| Systemic hormonal preparations, excl. sex hormones and insulins | 167 | 107 | $55.50 | $0.33 | $0.52 | 1,688 | 634 | $66,014.29 | $39.11 | $104.12 |
| Drugs for acid related disorders | 132 | 102 | $207.68 | $1.57 | $2.04 | 140 | 111 | $128.78 | $0.92 | $1.16 |
| Vitamin B12 and folic acid | 90 | 75 | $193.55 | $2.15 | $2.58 | 120 | 100 | $273.57 | $2.28 | $2.74 |
| Endocrine therapy | 18 | 12 | $298.54 | $16.59 | $24.88 | 710 | 428 | $17,114.61 | $24.11 | $39.99 |

All expenses were converted with annual average exchange rate (KRW/USD, see S2 Table).

the utilization of ART by women with infertility increased substantially among women in their late thirties in 2014 compared to 2008, with women aged 40 years utilizing ART most frequently in 2014 [25]. These results show an increasing trend in the age of patients seeking treatment for infertility, which are consistent with the current findings.

The number of male patients increased substantially, resulting in an increased proportion of male patients among the infertility patient population. According to the HIRA's Healthcare Big Data Disease Statistics, the number of male patients seeking care for infertility in South Korea increased progressively from 34,811 in 2010 to 52,902 in 2015, after which the number increased by larger increments to 61,903 in 2016 and 62,648 in 2017 [27]. This pattern has been interpreted by some as owing to the October 2015 update of the guidelines for infertility treatment that specified that "infertility of unknown cause" could be diagnosed only after male infertility testing and normal findings on semen testing, ovulatory function, and uterine-cavity and fallopian tube tests [28]. The number of male patients with infertility rose by 17% in 2016 compared to 2015, increased slightly, by 1%, in 2017, and again increased substantially, by

24%, to 77,971 in 2018, increases that could also be due to the higher number of men visiting a healthcare facility and being diagnosed with infertility to meet the eligibility criteria for ART health insurance coverage implemented in 2017. This might imply that health policies could affect the individual's intention to visit the healthcare facility, leading to delayed detection and disease diagnosis. The rate of childbirth management (Z31) increased markedly in 2018 compared to 2016. Speculation attributes this increase to the fact that the KCD code Z31 includes general counseling and recommendations for childbirth as well as ART (e.g., IVF, artificial insemination, and other assisted fertilization techniques) for both female and male patients. Z31 has been coded increasingly often since health insurance coverage began to include ART in 2017. Notably, acute bronchitis (J20), which was ranked third in comorbidities in both sexes in 2016, was not in the top 20 comorbidities associated with infertility in 2018. It is possible that acute bronchitis was an epidemic in 2016, or that changes in insurance coverage and policies contributed to fewer claims submitted with this code in 2018; this study cannot pinpoint the actual cause.

Less than 1% of women with infertility had diabetes mellitus, in both 2016 and 2018. A previous South Korean study that analyzed health insurance claims data showed that women with infertility who had been diagnosed with diabetes mellitus were less likely to report childbirth [22]. A Taiwanese study also showed that women with diabetes mellitus were at significantly lower odds for successful childbirth compared to their non-diabetic counterparts [29]. One difference between this study and previous studies is that previous studies set childbirth as the study outcome and examined factors that are significantly linked to this outcome; this study examined comorbidities in all patients who utilized healthcare for infertility regardless of their childbirth status.

A study that investigated infertility treatment provided by South Korean medicine doctors (KMDs) in South Korea reported that KMDs believed herbal medicine therapy to be the most effective KM treatment, both in the short term (3 months) and long term (1 year) [30]. However, herbal medicine—unlike acupuncture, moxibustion, and cupping therapy—is not covered by health insurance, so findings pertaining to KM utilization could be underestimated.

A slight increase in the number of patients with infertility from 2016 to 2018 was coupled with a more than four-fold increase in the cost of care for infertility, showing a markedly higher number of claims and expenses submitted for the treatments provided. The service code with the greatest increase in the number of claims and costs was treatment and surgery. These increases could be attributable to the introduction of ART-related medical practices, including embryo transfer through the cervix, embryo culture and observation, embryo screening and additional culturing, egg retrieval and treatment, and intrauterine insemination under treatment and surgery procedure codes for women by 2018, services that were not covered in 2016. The medication with the greatest increase in the number of claims and costs was gonadotropins. Under the gonadotropin category, various medications were introduced, most of them follicle-stimulating injections composed of menotropin or follitropin that were not covered in 2016. It can be inferred that launching health insurance coverage of ART substantially contributed to the elevated total cost of care for infertility.

The total annual cost of infertility treatment claimed for insurance reimbursement rose approximately 4.8-fold from 2016 to 2018; the per-patient annual expense increased approximately 4.5-fold during the same period. In the United States, infertility treatment is not covered by public health insurance. One US study estimated a couple's median total cost of infertility treatment over 18 months to be $5,338 (Interquartile range 1,197–19,840) [31]. When those costs are compared with the annual per-patient cost in 2018, as demonstrated in this study with its similar duration and population, the cost of infertility treatment seems to be considerably lower in South Korea than in the United States. Further, the American Society

for Reproductive Medicine reported the average cost of IVF per cycle to be $12,400 [32], while the average cost of an IVF cycle in South Korea before the launch of health insurance coverage in 2017 was 2.834 million KRW, excluding prematurely terminated cycles [28], which is approximately one-fifth of the total cost in the United States. This difference could be attributable to the study presenting only the costs submitted for insurance reimbursement (as opposed to the US study that summed the total cost of infertility treatment in the United States, where infertility procedures are not covered by public health insurance or most private health insurers). The difference could also be due to divergent calculation methods (mean vs. median) and the income gap between the two countries. The financial burden for infertility treatment and ART procedures varies across countries.

This study shows markedly increased number of claims and cost of care for infertility on the national scale. According to a 2017 Korean policy report [28], the total non-covered costs and total OOP costs for IVF since its inclusion in health insurance coverage in 2017 were approximately 1.408 million KRW, down to 40% compared to costs prior to IVF's inclusion in health insurance coverage. It is reasonable to say that national health insurance coverage of ART reduced the financial burden of individual patients with infertility.

In addition to the financial burden, treatment time and the difficulty of visiting healthcare facilities for care also pose social burdens. The average visits per patient increased by about 1 day, and the average visits per patient gap between the sexes widened in 2018; the average number of visits was markedly higher for women than for men, from 3.7 to 5.9 days. ART and other infertility treatments require frequent and sudden clinical visits depending on the menstruation cycle. Some women choose to quit their jobs until treatment yields success, despite the substantial cost incurred [33]. In May 2018, the South Korean government enacted a law to obligate employers to provide up to three days off from work per year when workers ask for a leave to seek infertility treatment (e.g., artificial insemination, IVF). This law is a measure to promote gender equality in employment as well as work-family balance. However, it is difficult for patients with infertility to utilize this leave since only the first day of the leave is paid and that payment is fully covered by the employer. The difficulty of requesting a work leave limits patients with infertility seeking medical care, highlighting the importance of both establishing a social system that is favorable toward infertility treatment and providing financial support for infertility treatment.

There were study limitations. First, as the 2016 and 2018 HIRA-NPS data provided by the HIRA were used, service codes not covered by insurance and indirect costs were not included in the analysis. Hence, it is possible that the number of patients seeking healthcare for infertility was underestimated. Second, this was a cross-sectional study investigating two years using HIRA-NPS data that is sampled annually from the entire population of South Korea (2016 and 2018). While healthcare services received during the corresponding year can be tracked, patients' data are only provided by one-year increments without continuity between each year. Thus, long-term follow-up of treatment was not possible using this data, and information on health-related outcomes of health insurance coverage of ART such as pregnancy success was not provided. Third, only patients aged 30 years or older were included; infertility treatments received by younger patients were not examined. This age criterion was set considering that the use of ART markedly increases in patients older than age 30 and that this age is gradually rising. However, this criterion limits the generalizability of the findings to the entire infertility population. Fourth, while childbirth can be used as an indicator for treatment success owing to the nature of infertility, only healthcare utilization was analyzed for infertility. Therefore, it could not be determined if health insurance coverage of ART actually increased the number of childbirths or increased the percentage of childbirths through ART in terms of total births.

Despite these limitations, this study demonstrated the following strengths. First, this study is the most recent to analyze healthcare utilization for infertility in South Korea before (2016) and after (2018) the implementation of health insurance coverage of ART (2017) comparatively. Second, accounting for the unique dual healthcare system in South Korea, this study examined KM utilization as well as WM utilization. Third, this study analyzed healthcare utilization for infertility by men and women separately. This prevents potential biases that can occur from sex-specific or mixed samples.

## Conclusion

As no previous study investigated infertility healthcare utilization changes since the launch of health insurance coverage of ART, this study is significant in revealing the effects of national health policies on the management, treatment, and cost of care of infertility. The percentage of patients aged 35 years or older and the proportion of male patients utilizing infertility healthcare increased. It was confirmed that the total cost of care for infertility increased substantially, and the number of medical procedures and the use of medications related to ART increased markedly. These findings could also be used for decision-making on insurance coverage of ART outside of South Korea. Subsequent studies should analyze the evolving health policies and healthcare utilization trends for infertility in the longer term, including childbirth outcomes, to assess the effectiveness of these policies.

## Supporting information

**S1 Table. Classification of medication.**
(DOCX)

**S2 Table. Annual average KRW-USD exchange rate and healthcare price index.** This information is available on the following website: Korean Statistical Information Service (http://kosis.kr); The price index represents the relative price level of cost adjusted as of 2018.
(DOCX)

**S3 Table. High-frequency comorbidities for female patients.** KCD: Korean Standard Classification of Diseases.
(DOCX)

**S4 Table. High-frequency comorbidities for male patients.** KCD: Korean Standard Classification of Diseases.
(DOCX)

**S5 Table. High-frequency medications for female patients.**
(DOCX)

**S6 Table. High-frequency medications for male patients.**
(DOCX)

**S1 Checklist. STROBE statement—Checklist of items that should be included in reports of *cross-sectional studies*.**
(DOC)

## Acknowledgments

No assistance in the preparation of this article is to be declared.

## Author Contributions

**Conceptualization:** Han-Sol Lee, Yu-Cheol Lim.

**Data curation:** Yu-Cheol Lim, Ye-Seul Lee.

**Formal analysis:** Yu-Cheol Lim, Ye-Seul Lee.

**Funding acquisition:** Dong-Il Kim.

**Investigation:** Han-Sol Lee, Yu-Cheol Lim.

**Methodology:** Yoon Jae Lee, In-Hyuk Ha, Ye-Seul Lee.

**Project administration:** Yoon Jae Lee, In-Hyuk Ha, Ye-Seul Lee.

**Supervision:** Yoon Jae Lee, In-Hyuk Ha, Ye-Seul Lee.

**Validation:** Dong-Il Kim, Kyoung-Sun Park.

**Writing – original draft:** Han-Sol Lee.

**Writing – review & editing:** Yu-Cheol Lim, Dong-Il Kim, Kyoung-Sun Park, Yoon Jae Lee, In-Hyuk Ha, Ye-Seul Lee.

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
