## [Decision Letter · Decision Letter 0]

25 Sep 2023

PONE-D-23-22683Comparative analysis of infertility healthcare utilization before and after insurance coverage of assisted reproductive technology: A cross-sectional study using National Patient Sample dataPLOS ONE

Dear Dr. Lee,

Thank you for submitting your manuscript to PLOS ONE. After careful consideration, we feel that it has merit but does not fully meet PLOS ONE’s publication criteria as it currently stands. Therefore, we invite you to submit a revised version of the manuscript that addresses the points raised during the review process.

We look forward to receiving your revised manuscript.

Kind regards,

Jae-Mahn Shim

Academic Editor

PLOS ONE

Journal Requirements:

**Additional Editor Comments:**

Both the reviewers requested minor revision. So, please respond to a few of their concerns in the reviewer comments.

Reviewers' comments:

Reviewer's Responses to Questions

**Comments to the Author**

1. Is the manuscript technically sound, and do the data support the conclusions?

Reviewer #1: Yes

Reviewer #2: Yes

2. Has the statistical analysis been performed appropriately and rigorously? 

Reviewer #1: Yes

Reviewer #2: Yes

3. Have the authors made all data underlying the findings in their manuscript fully available?

Reviewer #1: No

Reviewer #2: Yes

4. Is the manuscript presented in an intelligible fashion and written in standard English?

Reviewer #1: Yes

Reviewer #2: Yes

5. Review Comments to the Author

Reviewer #1: This is a descriptive analysis of infertility treatment in South Korea.

Please add the country name of "South Korea" in the abstract.

Please explain how the authors connected each treatment to the disease names of N97 or M46.

Reviewer #2: The topic of this thesis is a necessary topic in a medical environment where assisted reproductive technologies are often used to overcome low birth rates. It appears that appropriate analysis and conclusions were reasonably drawn using national statistics.

Is there any additional analysis of the impact of changes in health insurance coverage of ART on pregnancy success?

6. PLOS authors have the option to publish the peer review history of their article (what does this mean?). If published, this will include your full peer review and any attached files.

Reviewer #1: No

Reviewer #2: **Yes: **Hwang deoksang

---

## [Author Response · Author response to Decision Letter 0]

6 Nov 2023

Reviewer #1: 

1. This is a descriptive analysis of infertility treatment in South Korea.

Please add the country name of "South Korea" in the abstract.

- We appreciate the reviewer’s comment. Based on the comment, we revised the manuscript as follows:

This study aims to analyze the types and cost of infertility care provided in a clinical setting to examine the changes of healthcare utilization for infertility after the 2017 launch of assisted reproductive technology (ART) health insurance coverage in South Korea.

2. Please explain how the authors connected each treatment to the disease names of N97 or M46.

- We appreciate the reviewer’s comment. The treatments and disease names were connected if the claims record from the hospital/clinics included the disease codes of interest as the primary diagnosis or the main disease code. 

Based on the reviewer’s comment, we revised the Methods as follows:

Patients who received WM or KM services with International Classification of Diseases 10th revision (ICD-10) code N97 (female infertility) or N46 (male infertility) at least once in 2016 or 2018 were included in the study. The identified patients’ reimbursement claims data with N97 or N46 as the primary diagnosis were collected for the data analysis.

 

Reviewer #2: 

1. The topic of this thesis is a necessary topic in a medical environment where assisted reproductive technologies are often used to overcome low birth rates. It appears that appropriate analysis and conclusions were reasonably drawn using national statistics.

- We would like to thank the reviewer for his comments.

2. Is there any additional analysis of the impact of changes in health insurance coverage of ART on pregnancy success?

- We appreciate the reviewer’s comment. We agree with the reviewer that an important component in understanding the impact of health policy is the final health-related outcome, which in this case is the pregnancy success. To analyze pregnancy success, the data source should provide a minimum of 9 months’ reimbursement records of each patient who utilized ART. Unfortunately, the database used in this study did not provide longitudinal data of each patient but only provided one-year increments of patient data without continuity, and we were not able to complete long-term follow-ups.

Based on the reviewer’s comment, we revised the manuscript as follows:

Second, this was a cross-sectional study investigating two years using HIRA-NPS data that is sampled annually from the entire population of South Korea (2016 and 2018). While healthcare services received during the corresponding year can be tracked, patients’ data are only provided by one-year increments without continuity between each year. Thus, long-term follow-up of treatment was not possible using this data, and information on health-related outcomes of health insurance coverage of ART such as pregnancy success was not provided.

---

## [Editor Report · Decision Letter 1]

9 Nov 2023

Comparative analysis of infertility healthcare utilization before and after insurance coverage of assisted reproductive technology: A cross-sectional study using National Patient Sample data

PONE-D-23-22683R1

Dear Dr. Lee,

We’re pleased to inform you that your manuscript has been judged scientifically suitable for publication and will be formally accepted for publication once it meets all outstanding technical requirements.

Kind regards,

Jae-Mahn Shim

Academic Editor

PLOS ONE

Additional Editor Comments (optional):

Review comments/concerns from the previous review process have all been properly addressed in this revision.
---

## [Editor Report · Acceptance letter]

21 Nov 2023

PONE-D-23-22683R1 

Comparative analysis of infertility healthcare utilization before and after insurance coverage of assisted reproductive technology: A cross-sectional study using National Patient Sample data 

Dear Dr. Lee:

I'm pleased to inform you that your manuscript has been deemed suitable for publication in PLOS ONE. Congratulations! Your manuscript is now with our production department. 

Kind regards, 

on behalf of

Professor Jae-Mahn Shim 

Academic Editor

PLOS ONE